# Short-Term Evolutionary Features and Circadian Clock-Modulated Gene Expression Analysis of *Piezo*, *nanchung*, and *αTubulin at 67C* in a Romanian Population of *Drosophila suzukii*

**DOI:** 10.3390/insects16060591

**Published:** 2025-06-04

**Authors:** Adriana-Sebastiana Musca, Attila Cristian Ratiu, Adrian Ionascu, Nicoleta-Denisa Constantin, Marius Zahan

**Affiliations:** 1Faculty of Animal Science and Biotechnology, University of Agricultural Sciences and Veterinary Medicine of Cluj-Napoca, 400372 Cluj-Napoca, Romania; adriana-sebastiana.musca@usamvcluj.ro (A.-S.M.); mzahan@usamvcluj.ro (M.Z.); 2Life Sciences Institute “King Michael I of Romania”, University of Agricultural Sciences and Veterinary Medicine of Cluj-Napoca, 400372 Cluj-Napoca, Romania; 3Drosophila Laboratory, Department of Genetics, Faculty of Biology, University of Bucharest, 060101 Bucharest, Romania; a.ionascu20@s.bio.unibuc.ro (A.I.); constantin.nicoleta-denisa@s.bio.unibuc.ro (N.-D.C.); 4The Research Institute of the University of Bucharest, 050095 Bucharest, Romania

**Keywords:** *Drosophila suzukii*, short-term evolution, circadian clock, adaptability, gene expression

## Abstract

*Drosophila suzukii* is an invasive fruit fly species that causes major damage to crops. Its success is partly due to its ability to quickly adapt to new environments. In this study, we investigated two genes, *Piezo* and *nanchung* (*nan*), which are also influenced by the circadian rhythm. We analyzed their DNA sequences and gene expression in a Romanian population and compared them to those from the USA and Japan. Our findings show that these genes accumulate genetic changes in non-coding regions and respond quickly to changes in light cycles, with both genes being upregulated. This suggests that *D. suzukii* uses rapid genetic and physiological adjustments to cope with environmental stress.

## 1. Introduction

*Drosophila suzukii* was originally described by Matsumura in 1931 in Japan [1] and although the first isolated reports of damage caused by *D. suzukii* in Asia were made as early as 1939 [2], the presence of this pest in North America and Europe was only reported in the late 2000s, followed by a rapid expansion in these continents [3,4]. Unlike other *Drosophila* species, which infest decaying fruit, *D. suzukii* females use a serrated ovipositor to deposit their eggs in healthy, ripening fruit [5], making it vulnerable to secondary infections such as bacteria and yeasts. This damage can also increase the risk of other fly species attacking the affected fruit, which, along with the destructive action of larvae over the tissues, renders infested fruit unfit for sale [5,6,7].

The effective invasion of *D. suzukii* is at least partly explained by its remarkable adaptability to local conditions, thus assuring its survival and reproductive success. Frequently, the key environmental factors that vary across *D. suzukii*’s invasive range are the natural light–dark and temperature cycles, as well as humidity.

Prime candidate genes for studying adaptability traits are *nanchung* (*nan*) and *Piezo*, which play critical roles in developmental regulation and mechanosensation, respectively. The *nan* gene belongs to the Transient Receptor Potential (TRP) ion channel superfamily that is highly conserved from worms to flies to humans and impacts a plethora of behaviors and senses related to changes in environmental stimuli [8], including hygrosensation [9]. Based on evidence gathered from *Drosophila melanogaster* and other animal models, *D. suzukii nan* could engage in a phototransduction cascade and entrainment of the circadian clock [10], light discrimination, and color-driven behaviors [11], as well as in hearing [12]. *Piezo* is an evolutionarily conserved transmembrane protein involved in the perception of pressure and mechanical tension in mammals [13,14]. The expression of *Piezo* in the mechanosensory bristle neurons of male genitalia seems to promote reproductive success through the stabilization of copulation posture of *Drosophila* males [15]. It is also associated with several important processes, such as the generation of satiety signals [16] and sleep onset latency, a trait associated with the circadian clock [17]. Also, *Piezo* could play a role in the correct expression of ion channels [18] and thus collaborate with *nan* to assure mechanosensitivity, a trait relevant for sensing more specific ranges of substrate stiffness [19], which impacts choosing the optimal substrates for egg laying [20].

In *Drosophila*, both *Piezo* and *nan* are expressed in a specific class of sensory structures named chordotonal organs [15,21,22] and through them are involved in proprioception and locomotion [23], as well as substrate-born communication, a natural mode of signal transfer extensively used in the wild [24,25]. Moreover, chordotonal organs are essential for synchronizing the circadian clock with daily temperature changes [26]. The circadian rhythms regulate numerous biological processes by tuning them in order to occur at beneficial times, a feature that is possible due to synchronization with the environment by so-called “Zeitgebers” [27]. Two crucial Zeitgebers are the light–dark ratio and ambient temperature, which are conditioned by region-specific diurnal cycles. The roles of *Piezo* and *nan* in temperature synchronization, light and color-related behaviors, sleep latency, and thermal plasticity suggest they could be instrumental in modulating the circadian cycle, enabling *D. suzukii* to optimize its behavior and success in diverse climates.

Understanding the genetic basis of *D. suzukii*’s adaptability through the lenses of *Piezo* and *nan* genes specific structural evolution in various natural populations, as well as their reactivity to impromptu diurnal changes, has significant implications for agroeconomics and ecology. From an agroeconomic perspective, this knowledge can provide information on the development of targeted pest control strategies. Additionally, identifying genetic markers associated with adaptability could aid in monitoring and predicting the species’ range expansion, enabling proactive management measures.

Herein, we aimed to perform two distinct investigations regarding *Piezo* and *nan* genes, particularly in individuals pertaining to the ICDPP-ams-1 line derived from a Romanian natural population of *D. suzukii* that was recently sequenced and has its genome assembled at the contig level [28]. One objective consisted of analyzing the short-term evolution of the mutational landscape of *Piezo* and *nan* by comparing specific sequence data retrieved from ICDPP-ams-1 genome assembly with data available from Dsuz_RU_1.0 and CBGP_Dsuzu_IsoJpt1.0, the former and, respectively, current reference genomes of *D. suzukii*. In addition, we investigated whether artificially controlled modifications of photoperiodicity enable effective modifications of *Piezo* and *nan* gene expression in ICDPP-ams-1 female adult individuals. In addition, we also used *αTubulin at 67C* (*αTub*), a housekeeping gene evolutionarily conserved from yeast to *Drosophila* to human [29], as a reference for DNA variants evaluation and phylogenetic analysis, as well as for normalization in the context of gene expression analysis.

## 2. Materials and Methods

### 2.1. D. suzukii Husbandry

The *D. suzukii* individuals used in this study originated from the ICDPP-ams-1 line derived from a Romanian local population, as previously described [28].

Flies were reared in 50 mL culture bottles, with a base layer of approximately 10–15 mL of standardized culture medium. This medium was prepared according to an established protocol and consisted of baker’s yeast, banana, sugar, wheat semolina, bacteriological agar, and distilled water [28]. Propionic acid was added to prevent contamination by unwanted microorganisms.

### 2.2. Circadian Rhythms Experimentation

The adults subjected to circadian rhythm changes stress were actually the F2 descendants of adult flies transferred from a natural circadian cycle to a controlled 12 h:12 h light–dark (LD) cycle, with the light phase synchronized to the 08:00–20:00 interval. This transition was achieved by placing vials with varying numbers of adult *D. suzukii* in a separate environmental chamber maintained at 22 °C under the specified light–dark cycle. Following pupal emergence, adult flies were removed, and the newly emerged F1 generation was maintained under identical conditions. This process was repeated for an additional generation, which was subsequently used for experimentation. At one day post-emergence, the F2 individuals were sexed and separated accordingly.

The collected F2 individuals were then assigned to three experimental groups: the control group, experimental group A, and experimental group B, each group consisting of three biological replicates (BR). Since the experiment was conducted twice, the number of individuals in each BR varied between experiments. In the first experiment, each BR consisted of 20 females and 20 males, whereas in the second experiment, each BR contained 19–20 females and 11–20 males.

Individuals in group A were maintained on a 12 h:12 h LD cycle for eight days. Afterward, the LD cycle was shifted to 16 h:8 h, where they remained for five days before being collected for RNA extraction. Similarly, individuals in group B were maintained for eight days on a 12 h:12 h LD cycle. Following this, the LD cycle was changed to 16 h:8 h for three days, after which they were returned to a 12 h:12 h LD cycle for an additional two days before being collected for RNA extraction. The complete experiment design is illustrated in Figure 1.

### 2.3. Incubator Design

To maintain stable temperature and light conditions during the experiment, an original controlled environmental system in the form of a custom incubator was constructed (Figure 2). This system consisted of a chamber with a flexible lid (1), providing space for vials containing *D. suzukii* individuals.

LD cycles were regulated via timer-controlled electrical outlets (3) connected to LED light sources. To minimize temperature fluctuations, a heating plate (5) was installed at the base of the incubator and linked to a microcomputer with a temperature controller (2). The controller featured a temperature sensor (6) that continuously monitored internal conditions and activated the heating system as needed. To prevent direct overheating of the vials, they were placed on a metal mesh (4) positioned 3 cm above the heating plate. The incubator measured 50 cm in length, 44 cm in width, and 40 cm in height.

### 2.4. RNA Extraction and Quantitative Real-Time PCR (qRT-PCR) Reactions

From each BR, 7–17 females were collected, placed in tubes, and gently ground, after which 200 µL of RNAlater (Invitrogen, Thermo Fisher Scientific, Waltham, MA, USA, ref. AM7024) was added. The samples were stored at −80 °C until RNA extraction, which was performed using the Monarch Total RNA Miniprep Kit (New England Biolabs, Ipswich, MA, USA, cat. no. #T2110).

Reverse transcription was conducted using the ProtoScript II First Strand cDNA Synthesis Kit (New England Biolabs, cat. no. E6560), allowing the conversion of 1000 ng of RNA into the corresponding cDNA.

For qRT-PCR reactions, three BR were used, each analyzed by using three technical replicates. qRT-PCR was performed using the Luna qPCR Kit (New England Biolabs, cat. no. E3005), with *αTub* as the housekeeping gene for normalization. The primer sequences for *αTub* (forward: AGGATGCGGCGAATAACT; reverse: CGGTGGATAGTCGCTCAA) were obtained from published data [30], whilst primers for *Piezo* (forward: AGGATGCGGTTACGGTGTTG; reverse: CGGCTCCTCTAGAGGCCAGG) and *nan* (forward: GGTGATCGTGAGTGTGTTGC; reverse: CGGCCATCTGACATTGACTG) were manually designed. The *αTub* gene was chosen as housekeeping thanks to its proven stable expression in biotic and abiotic conditions (developmental stage, tissue, population, photoperiod, and temperature) [30]. Each qPCR reaction contained 1X qPCR Luna master mix, 20 ng of cDNA, and 0.16 µM of forward and reverse primers. The Ct values were analyzed using the qDATA application [31].

### 2.5. Evolutionary Changes and Phylogenetic Analyses

In order to perform the evolutionary analysis, we used the ICDPP-ams-1 (GenBank: GCA_040114545.1), Dsuz_RU_1.0 (GenBank: GCA_037355615.1), and CBGP_Dsuzu_IsoJpt1.0 (GenBank: GCA_043229965.1) genome assembly data available in the NCBI database. Dsuz_RU_1.0 and CBGP_Dsuzu_IsoJpt1.0 are the previous and, respectively, current reference assemblies, both of them consisting of assembled chromosomes/scaffolds and having extensive gene, transcript, and protein annotations. Of note, Dsuz_RU_1.0 corresponds to a population found in the United States of America (USA), and CBGP_Dsuzu_IsoJpt1.0 to a Japanese population. The ICDPP-ams-1 assembly comprises a collection of approximately 6700 unordered contigs.

Gene sequences for *Piezo*, *nan*, and *αTub*, along with their corresponding transcript data, were directly downloaded from NCBI for the reference genomes. For ICDPP-ams-1, to obtain the gene sequences, we first performed a minimap2 [32] assembly of the contig collection versus (vs.) the chromosomes of Dsuz_RU_1.0. We did not use the current CBGP_Dsuzu_IsoJpt1.0 reference because we noticed serious problems regarding their third chromosome assembly, thus preferring the prior reference. The minimap2 [32] assembly data were interrogated using the genomic coordinates of *Piezo*, *nan*, and *αTub* from Dsuz_RU_1.0, thus allowing the extraction of the corresponding assembly data from ICDPP-ams-1.

The consensus sequences retrieved from ICDPP-ams-1, together with the references and downloaded data, were submitted to multiple alignment using Clustal Omega 1.2.2 [33,34] and Muscle 5.1 [35] heuristics. To identify exonic and intronic conflictual coordinates harboring differences between gene sequence data pertaining to the three compared genomes, we used for multiple alignment both gene and transcript sequences (available for reference data). The results were exported as .csv files and then manually analyzed. The drawing of phylogenetic trees with PhyML 3.0 software [36] used alignment data generated exclusively using the gene sequences.

The minimap2 [32] mapping and ICDPP-ams-1 gene data retrieval, the multiple alignments procedures and their graphical representation, the marking of conflictual coordinates and the generation of corresponding .csv files, and the drawings and analysis of the phylogenetic trees were performed using the Geneious Prime 2025.0.3 application (Geneious Prime 2025.0.3).

We used FlyBase [37] (release FB2025_01) to find information on phenotypes and references regarding *nan*, *Piezo*, and *αTub* orthologous genes from *D. melanogaster*. From within the same database, we also obtained the sequences of the three genes according to Release 6.63 of the *D. melanogaster* reference genome. The respective gene sequences were multiple-aligned with their corresponding *D. suzukii* orthologs using the Geneious Alignment implicit option, and then phylogenetic analysis was performed with PhyML 3.0 software [36].

### 2.6. Biostatistics

The observed vs. expected distribution of conflictual coordinates within exonic and intronic regions corresponding to *Piezo*, *nan*, and *αTub* genes from the three genome assemblies was analyzed with Fischer’s exact test executed by GraphPad Prism 8.4.2 (GraphPad Prism version 8.4.2 for Windows, GraphPad Software, Boston, MA, USA, www.graphpad.com).

The nonparametric Mann–Whitney U testing was automatically performed when running the integrated qDATA procedure for analyzing Ct data gathered from qRT-PCR experiments [31].

## 3. Results

The results generated within our study cover a merge of structural and functional genomics analyses aiming to evaluate short-term evolutionary changes in *Piezo*, *nan*, and *αTub* genes from three *D. suzukii* populations, as well as the impact of photoperiodicity changes on the expression of genes linked to the circadian cycle. For the latter endeavor, we designed an original experimental setup for batches of *D. suzukii* individuals pertaining to ICDPP-ams-1 line derived from a local Romanian population. The relative gene expression of *Piezo* and *nan* was assessed in females collected at the end of two distinct experimental runs.

### 3.1. Phylogenetic Analysis

In order to obtain the sequence corresponding to *Piezo*, *nan*, and *αTub* genes from the ICDPP-ams-1 line, we followed a multi-step approach. Firstly, the complete list of contigs corresponding to the ICDPP-ams-1 was assembled with minimap2 to Dsuz_RU_1.0, the former reference genome for *D. suzukii*. Then, we retrieved the genomic location of the three genes within Dsuz_RU_1.0 from the publicly available genome annotation table, and based on these coordinates, we extracted the corresponding regions from the minimap2 assembly. Since the considered list of contigs is by itself the result of an assembly procedure, we expected a relatively low genomic coverage, especially within regions overlapping with gene content. For our selections, the actual assembly coverage ranged between one and two; thus, for a given gene, the subsequences with a coverage of one are identical to the respective contig sequence. When two overlapping contigs were present, an automatic consensus sequence was inferred, while a few contig sequence conflicts were manually resolved. In rare cases where one contig had an identic nucleotide to the reference and the other contig harbored a different one at the same coordinate, we opted to retain the reference one. Because of this strategy, it is possible that we might have missed or added some single nucleotide differences. Each asserted gene sequence was put together with the corresponding gene and primary transcript sequences from both Dsuz_RU_1.0 and CBGP_Dsuzu_IsoJpt1.0 genome assemblies, and the resulting sets of five sequences were submitted to multiple alignment by implementing either Clustal Omega for *Piezo* or Muscle for *nan* and *αTub*. When one uses both the gene and their corresponding transcript sequences for multiple alignment purposes, the specific heuristic is likely to correctly detect the complete local multiple alignments comprising subsequences from all the analyzed sequences. Alternatively, if only some of the sequences are represented in an incomplete local multiple alignment, then the respective sectors of the other sequences will be represented by gaps. Owing to this particularity, we were able to highlight the gene subsequences pertaining to exons and introns, each exposed by complete or, respectively, incomplete local multiple alignments. In our experience, the most efficient heuristic in terms of precisely delimiting the exonic and intronic regions was Muscle, but it failed to run on the *Piezo* set because of this gene extensive length and our computational limitations. Although it failed to provide a reliable multiple alignment for *nan*, Clustal Omega, which uses significantly fewer computational resources than Muscle, was very effective on *Piezo*. In addition, within the Geneious Prime output, the conflicts (differences) between the sequences forming complete or incomplete local multiple alignments are implicitly emphasized. Considering our results, if we focus on a particular position or subsequence within a gene, that specific nucleotide or group of nucleotides is covered by at least one and a maximum of five sequences from the multiple alignment. The local multiple alignments that have a coverage of one or two are marked as conflictual. Similarly, when specific local multiple alignments covered by three to five sequences harbor single or consecutive differences between at least two of them, the respective coordinate/subregion is also marked as conflictual. These conflicts are highlighted by colored nucleotides or, when zooming out the graphical representation, by black lines or rectangles (Figure 3).

We generated .csv files containing the conflictual coordinates and exploited them in order to evaluate the gene subregions amenable to accumulating mutations in the short time span since the three populations diverged from the original one, alleged evolved in Japan. Regardless of the isolated or grouped coordinates were highlighted as conflictual, we individually considered each marked coordinate and ascribed it to exonic (including UTR exons) and intronic regions and ran Fisher’s exact test to evaluate the discrepancy between the observed and expected distribution of mutated positions within the respective gene regions (Table 1). We considered that it was appropriate to stress that within a certain multiple alignment, as a specific coordinate could harbor either single or multiple variants, but it was counted just once. Also, for the *Piezo* gene, we assumed two counting strategies: one considering its entire length and the other excluding its first 5’UTR intron that represents about 55% of the total intron length and was expected to harbor a significant proportion of intronic conflictual coordinates. Both graphical visualization (Figure 3) and statistics showed that the number of coordinates with various mutational loads of intronic regions is significantly greater than that attributed to exons. Based on our data, we considered an original parameter we named mutation accumulation tendency (MAT), representing the ratio between the fraction of conflictual single coordinates found within introns and the fraction of total gene length occupied by introns (Table 1). We found that a greater MAT value is indicative of genes harboring an excess of intronic mutations, a feature that can be indicative of the evolutionary shielding of the actual coding regions.

As somehow expected, since *αTub* is a housekeeping gene, it has the highest MAT score, in contrast with *Piezo*, which has the most extensive cumulative intronic region that can easily fit a significant proportion of conflictual coordinates.

In order to draw phylogenetic trees representative of complete genes, we separately performed another round of multiple alignments using the same heuristics but excluding the transcripts. Based on the resulting multiple alignments, the actual trees were built by using the PhyML 3.0 software that uses maximum-likelihood methods of phylogenetic inference (Figure 4), an approach viewed to be better than other alternative methods, such as distance or parsimony [36]. For the actual run, we opted for the Tamura–Nei substitution model, which estimates the transitional and transversional substitutions per site, as well as their total, taking into account excess transitions, unequal nucleotide frequencies, and the variation in substitution rate [38].

The results analysis emphasizes that the apparent phylogenetic distances between the variants of any specific gene have smaller values when the Romanian local population (ICDPP-ams-1) is compared to either the USA population (Dsuz_RU_1.0) or the Japanese population (CBGP_Dsuzu_IsoJpt1.0) than that inferred when comparing the USA and Japanese populations. Furthermore, there are fewer differences between ICDPP-ams-1 and Dsuz_RU_1.0 than between ICDPP-ams-1 and CBGP_Dsuzu_IsoJpt1.0. By short-term evolution, we named the gene structural modifications that occurred between the considered populations during a relatively small temporal interval. From the currently available evidence, the worldwide invasion of *D. suzukii* started approximately a century ago; therefore, this interval could be considered as the maximum period length for evolving noticeable differences between original populations and the migrating ones. In order to evaluate the long-term impact of the accumulation of genomic structural differences, we compared the gene sequences from the considered *D. suzukii* populations with those pertaining to their counterparts obtained from the *D. melanogaster* reference genome. The phylogenetic branch lengths separating *D. melanogaster* from the three *D. suzukii* populations were estimated based on the differences between the *Piezo*, *nan*, and *αTub* genes in the four genomes and have the following values: 0.2163, 0.2205, and 0.2273, respectively. The mean phylogenetic distances between the *D. suzukii* populations for *Piezo*, *nan*, and *αTub* are 0.016, 0.0245, and 0.0265, respectively, almost ten-fold smaller than the distances calculated when *D. melanogaster* was considered.

### 3.2. Gene Expression Analysis

Relative gene expression of *Piezo* and *nan* in *D. suzukii* females from the ICDPP-ams-1 line is varying consecutive to changes in the regular circadian clock, consisting of an unabated 12 h:12 h LD cycle applied to the control group. Experimental group A was subjected to an alteration of the regular circadian cycle after eight days to a 16 h:8 h LD routine, whilst for the experimental group B we modeled a series of changes, a first switch after eight days to the 16 h:8 h LD timetable, and three days later by a reversal to the regular cycle.

The gene expression assessment was accomplished on two distinct experiments, denoted 1 and 2, using similar numbers of individuals, with a slightly decreased number of males per BR in the second one. The complete Ct data for a given control or experimental group consisted of nine values equally distributed within the three BRs. Data exploitation made use of a version of Livak calculations [39] that rely on generating all the possible ∆Ct within the considered BR and was performed with the qDATA application [31]. For both genes, the results were approximately consistent between the two experiments in group A, but were conflicting within group B. Both experimental groups were being evaluated against the control one. Closer inspection of the Ct data revealed two categories of apparently spoiled values, most probably because of some unaccounted technical issues. One category of flawed data consisted of complete BRs that greatly deviated from their accompanying BRs, and the second error source involved single outlier Ct values. In order to minimize the impact of these problematic values and not wanting to exclude complete BRs, we decided to employ the mean Ct values of the eccentric BRs and to omit the outlier Ct values, respectively. Thus, new datasets containing various modifications relative to the initial data were used for relative gene expression estimations for either the single experiments or their aggregate values by using qDATA [31]. Although the results presented in Table 2 display notable differences between the logarithmic fold change (log_2_FC) values corresponding to the two experiments, the overall trend of gene expression fluctuations observed in group A and group B is upheld for both genes. The same tendency was spotted when using the cumulative data. More specifically, both genes are overexpressed consecutive to changing the circadian clock in groups A, but the gene expression levels are reduced in groups B, after resetting the photoperiodicity to regular intervals. Moreover, the amplitude of relative gene expression log_2_FC values was increased for group A when the cumulative data were used compared to individual experiments. However, cumulative data for group B did not accurately reflect the trend observed in individual experiments. Namely, data corresponding to the first experiment generated a high negative log_2_FC value, whilst second experimental data generated a positive value, but the combined data generated a positive value for both genes. This observation might indicate that using the cumulative data generates more robust results due to a degree of data normalization when using more values for both the experimental group and the control group.

Statistical analysis of 2^−ΔCt^ values corresponding to the experimental groups was performed using the nonparametric Mann–Whitney U test, as the majority of values were not normally distributed. In experiment 1, all group comparisons (A vs. control and B vs. control) revealed statistical significance for both genes, except for *nan* when comparing A vs. the control groups (*p* = 0.271). A similar trend was observed in experiment 2, where only data corresponding to *nan* when comparing B vs. control groups (*p* = 0.324) did not show statistical significance. Interestingly, using the cumulative data from both experiments revealed statistical significance in the case of *Piezo* when comparing A vs. the control groups (*p* = 9.105 × 10^−5^). Similarly, statistical significance was observed for both *Piezo* and *nan* genes when comparing A vs. the control groups. The comparison of B vs. the control groups was not statistically significant for either *Piezo* (*p* = 0.784) or *nan* (*p* = 0.168). All these statistical results are summarized in Table 2.

Additionally, data analysis revealed an overall downregulation of *Piezo* and *nan* genes in the B groups compared to the A groups. For *Piezo*, data from experiment 1 generated a −2.184 log_2_FC value and data from experiment 2 generated a 0.084 log_2_FC value. The trend for *nan* was more obvious, with −1.735 log_2_FC from experiment 1 and −1.176 log_2_FC from experiment 2. Except for the case of *Piezo* when comparing B2 vs. A2 groups (*p* = 0.097), all gene expression differences were statistically significant. When considering the cumulative datasets, the log_2_FC values calculated for *Piezo* and *nan* were −0.387 and −0.894, respectively, both being highly statistically significant (Figure 5).

## 4. Discussion

Since the focus of our study is the Romanian local population of *D. suzukii*, we evaluated the nucleotide variants (SNPs and indels) particular to this population in contrast with those found in reference genomes representing the USA and Japanese natural populations. The appraisal of these differences was performed for the two genes of interest, *Piezo* and *nan*, and *αTub*.

In *D. melanogaster*, only singular insertional alleles of *Piezo* [40,41] and *nan* [42], allowing tissue-guided expression with the use of GAL4 drivers, can determine semi-lethality or lethality, respectively. Otherwise, according to FlyBase [37], most mutant alleles of these two genes are viable. Regarding *αTub*, there are several studies describing alleles that lead to complete embryonic lethality [43,44,45,46]. Therefore, we consider that *Piezo* and *nan* cannot be cataloged as essential genes, although the latter could be involved in more critical biological processes when counting the phenotypic severity determined by its alleles, while *αTub* has all the traits classifying it as an essential housekeeping gene. Moreover, *αTub* is highly conserved through the living world, being ascribed to the housekeeping genes category starting from yeast and microscopic *Trichoderma* fungi [47] to insects such as cotton leafhopper, a severe pest of cotton and okra [48], and *D. melanogaster* [29].

The distinction between housekeeping genes and other gene categories is of interest for both translational and basic research. The term housekeeping gene is somewhat elusive, and its comparison with that of essential genes can be misleading [49]. In summation, a housekeeping gene is defined as a gene that is consistently expressed across tissues and developmental stages, being frequently associated with basic cellular maintenance pathways and evolutionary conserved [50,51]. The essential genes are believed to be indispensable for the survival of an organism, a property frequently tested with their corresponding null alleles. Since many housekeeping genes can determine lethal phenotypes through their mutant alleles, researchers are sometimes erroneously substituting housekeeping for essential genes in evolutionary studies and consider them to have the same evolutionary conservation [49,52,53].

It was previously demonstrated that in human and other animal species that housekeeping genes express more evolutionary conservation than other genes, either when compared to disease and non-disease genes [52,53] or to sets of essential genes [49]. Such comparisons can address long-term and/or short-term evolutionary characteristics as for instance the evolutionary rate or SNP density, respectively. The migration of *D. suzukii* started most probably in Japan during the first half of the 20th century, and it took about a century for this species to reach and colonize first the USA and then Europe. Consequently, we reasoned that it is more appropriate to evaluate short-term evolutionary differences through the density analysis of SNP and indel variants within both exons (including UTR exons) and introns (including UTR introns). Our analysis showed that the majority of single genomic coordinates that may harbor specific nucleotide variants are placed within introns, with *Piezo* having a maximum percentage of such conflictual coordinates. We applied a straightforward contingency analysis in order to compare the actual distribution of conflictual coordinates with a theoretical one that might have been achieved if they were randomly distributed within the gene, and the results were highly statistically significant. Accordingly, the evidence pointed to an evolutionary tendency for each of these genes to house mutations, especially within intronic regions. To the best of our knowledge, this study is the first report of such a trend in *Drosophila* genomes. Actually, this very subject was only recently addressed in humans, and the fact that the introns appear to be under weaker selection than exons was noted for somatic [54] and not germline [55] cells. This could be determined by a mismatch repair system that is available only for exons and may be correlated with their methylation status, a phenomenon more common in exons compared to introns [56]. Additionally, the tendency of mutations to be identified predominantly in introns, compared to exons, could be expected due to a form of survival bias. As mutations in introns are expected to be less damaging, they may have a greater sequence diversity compared to exons.

In light of our results, when the introns represent a large fraction of the gene, there is a high probability that most variants are found within their borders. Nonetheless, only a small intron fraction that contains the majority of variants shows a strong tendency for safeguarding the integrity of exon sequences in exchange for that of introns. We calculated the so-called MAT value that can indicate how strong the bias is toward retaining specific variants within intronic regions, a trend more evident in the case of evolutionarily conserved genes. Here, we calculated the highest MAT value for *αTub*, which is only slightly bigger than the MAT of *nan. Piezo*, regardless of whether we considered or not the extremely large intron of its 5’UTR region, exhibited low MAT scores. These results could hint at differences regarding the functional ranking of the two target genes. Indeed, *nan* is associated with highly conserved functions impacting neural pathways and various fundamental behaviors, while *Piezo* is mainly involved in useful but relatively ordinary processes.

To detect the genetic mechanisms associated with the successful invasion of *D. suzukii*, Olazcuaga et al. (2020) used 22 samples from different native and invasive populations [57]. The study found that a relatively small number of 169 SNPs can be correlated with the invasiveness of *D. suzukii*, most of them being singularly located within or nearby a total of 130 genes. Interestingly, long non-coding RNAs represented over 10% of the genes identified, highlighting the essential role of genetic regulation in the short-term adaptive response [57]. A study by Feng et al. (2024) analyzed genomic data from 29 *D. suzukii* samples collected from both native and invaded areas from East Asia, Hawaii, the Americas, and Europe [58]. They observed a significant influence of repetitive elements on the divergence of non-coding sequences, a feature indicating that the adaptation of nomadic species could be mainly determined by changes in gene regulatory mechanisms, rather than direct alterations of the encoded proteins [58]. Intronic sequences are known to harbor a vast variety of sequence features enabling complex patterns of gene regulation, therefore, the accumulation of mutations within their confines should also have a finite evolutionary tolerance [56]. Our study demonstrated a similar trend for the studied genes that contain variable genomic sites, especially within introns, but we did not go further with a more formal analysis. Based on individual gene sequence divergence estimated for *nan*, *Piezo*, and *αTub* between the three *D. suzukii* populations, we traced a draft evolutionary history. Thus, the results indicate that the Romanian population is marginally more related to the USA population, probably because the most plausible expansion route in Europe was through the USA. The apparent phylogenetic distance between the USA and Japanese populations was larger than the distances between the Romanian population and either of the other two, a feature that can indicate that the respective populations maintained their reproductive isolation and had parallel evolutionary histories. These features were scored for all three genes, with the maximum phylogenetic distance of approximately 0.0533 being calculated for *nan* between ICDPP-ams-1 and CBGP_Dsuzu_IsoJpt1.0 populations. To also assess how long-term evolutionary separation impacts the accumulation of genomic sequence small variants, we compared *Piezo*, *nan*, and *αTub* genes from the three *D. suzukii* populations with their orthologous counterparts from *D. melanogaster*. By some estimates, the separation between *D. suzukii* and *D. melanogaster* occurred about 9 to 12 million years ago [59]; thus, the considerable interspecies differences as opposed to intraspecies ones.

Regarding the stress induced by changes in the circadian clock, our approach considered the use of female individuals. The rationale for our choice stems from several considerations. Thus, we are aware that historically, few circadian studies included females, mainly because it is largely accepted that males are more responsive to changes in light conditions [60,61]. Interestingly, the same studies stress that this lack of data concerning female reactions to light cycle perturbations should be thoroughly addressed. Moreover, it seems that the female circadian system is less affected by perturbations in clock neurons; therefore, their circadian timekeeping is more robust [61]. Considering these, we believe that selecting females for gene expression analysis is of relevance because of the contribution to a topic marked by data scarcity regarding females. Also, gene expression modifications that are relevant in females, which are less sensitive to light-related disturbances, are very likely to be mirrored in males.

The experimental results indicate that the gene expressions of *Piezo* and *nan* are significantly influenced. One batch of flies (consisting of groups A1 and A2) was exposed to a single change in photoperiodicity and consequently showed an increased expression of these genes as compared to the control group, suggesting an adaptive response to LD cycle disruptions. On the other hand, the flies comprising the B1 and B2 groups were subjected to an initial change in the LD ratio, followed by reinstating the initial photoperiodicity. Our calculations showed a reduction in gene expression in comparison to group A, but still increased in rapport to the control (but not statistically significant), indicating a possible mechanism of readaptation. Considering the limited number of days spanning the periods with affected photoperiodicity, the rapid changes in gene expression profiles are quite remarkable. Although the results collected from the two distinct experiments are contradictory to some extent, regardless of whether *nan* or *Piezo* was considered, we observed a common pattern. More precisely, we distinguished a significant increase in gene expression in group A coupled with a significant decrease in gene expression in group B, the latter reaching sometimes values similar to those particular for the control group. Of note, the upregulation of *Piezo* and *nan* seems to be proportional to the number of days spent under increased light exposure (16 h:8 h LD cycle).

Our data suggest a light-dependent regulation of both *Piezo* and *nan*, in accordance with the known light sensitivity of TRP channels [8,10,62], as increased light exposure might result in increased *nan* gene expression. In this context, TRP channels are involved in light-dependent movement via Ca^2+^ dependent interactions [8,10,63]. Interestingly, the circadian clock has been documented to influence *D. melanogaster* preference towards green and red light based on TRP channels interactions [11]. Our experimental setup made use of LED light, that is a common source of excess blue light, which might have been a stress factor on the circadian rhythm, which, in turn, may cause tissular inflammation [64,65,66]. Inflammatory signals potentiate *Piezo* channels mediated by Ca^2+^ dependent interactions [67]. The involvement of both genes in the sleep behavior [17,68] might indicate a common biochemical pathway that could synergistically enhance their gene expression. Taken together, increased *nan* expression could be caused by the prolonged light cycles, whilst *Piezo* upregulation might be indirectly influenced following TRP channels overstimulation and *nan* overexpression. In support, the fly’s behavior and environment might affect the circadian clock by sensory activation [69,70,71].

Several studies have shown that urban populations of *D. suzukii* adapted their circadian rhythm regulation to better tolerate artificial light at night (ALAN), in contrast to rural populations, where these disturbances are more severe [72]. The adaptations include upregulation of the *LARK* gene, an important circadian regulator, but may also involve genes responsible for mechanical and auditory perception, such as *Piezo* and *nan*, thus contributing to the ecological success of the species [72].

TRP channels are also involved in the regulation of circadian behavioral rhythms, such as thermal preferences and activity cycles [73]. Furthermore, photoreceptors and TRP channels can function independently to regulate the circadian rhythm through their own photoreception mechanisms, in which these channels play a central role, being important in integrating environmental factors to optimize sleep–wake cycles [74]. Thus, considering that the human extensive transport system could rapidly change the location of smaller or larger groups of *D. suzukii*, the change–response period should be as short as possible in order to assure the survival.

Our results seem to indicate such fast response mechanisms, a genomic feature that was previously demonstrated also for other genes associated with circadian rhythm, such as *timeless* and *clock*, in response to temperature variation challenges [75,76].

## 5. Conclusions

The results obtained in this study highlight the evolutionary and functional dynamics of *Piezo* and *nan* genes in adapting *D. suzukii* populations. Comparative analysis of gene sequences allowed the identification of SNP and indel variants specific to the *D. suzukii* population in Romania, compared to reference genomes from the USA and Japan. A greater evolutionary proximity was confirmed between the ICDPP-ams-1 line and the Dsuz_RU_1.0 genome, compared to CBGP_Dsuzu_IsoJpt1.0, which may reflect a common geographical origin or a more recent separation between these populations. The mutations were distributed predominantly in intronic regions, an aspect supported by the high values of the MAT parameter, which suggests an evolutionary conservation of coding regions.

From the perspective of gene expression, the results demonstrated that changes in the circadian rhythm significantly influence the transcription of *Piezo* and *nan* genes. Both were overexpressed under conditions of a prolonged exposure to light (16 h:8 h LD cycle for group A), and their expression was reduced following the return to the standard regime (12 h:12 h LD cycle for group B). These fluctuations were supported by consistent log_2_FC values and relevant statistical significance, especially for group A, indicating a robust transcriptional response to changes in photoperiodicity.

The integrated interpretation of structural and functional data indicates that the studied genes not only accumulate genetic differences in non-coding regions, but also respond through expression changes to external environmental perturbations, such as circadian rhythm variations. These observations support the hypothesis of an active molecular adaptation to new environmental conditions that contributes to the invasive success of *D. suzukii*. In this context, the *Piezo* and *nan* genes may be relevant markers for investigating the molecular mechanisms involved in the circadian stress response and the ecological adaptation process.

## Figures and Tables

**Figure 1 insects-16-00591-f001:**
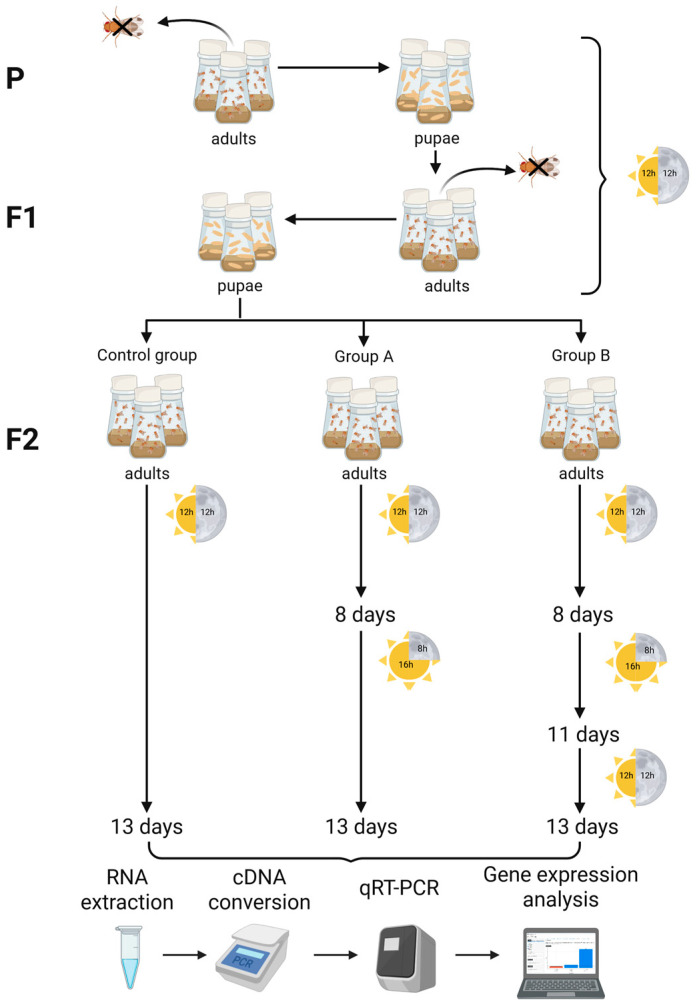
Graphical summary of the experiment investigating the influence of the LD cycle on the expression of *Piezo* and *nan* genes. P and F1 indicate the parental and, respectively, their first-generation descendants that were constantly exposed to a 12 h:12 h LD cycle. A selection of F1 descendants (F2) was subjected for a period of 13 days to either the same LD cycle as their parents or to experimental modifications to the LD cycle, supposing a single or two LD cycle switches after an initial eight-day interval of a 12 h:12 h LD cycle. After the termination of experimental monitoring, we processed genomic DNA material from females and performed qRT-PCR analysis on target genes. Created in BioRender.com (accessed on 16 April 2025).

**Figure 2 insects-16-00591-f002:**
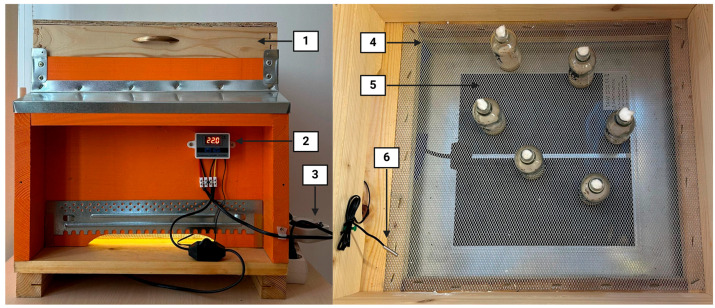
The original incubator built for this experiment includes (1) a flexible lid, (2) a temperature controller, (3) timer-controlled electrical outlets, (4) a metal mesh, (5) a heating plate, and (6) a temperature sensor.

**Figure 3 insects-16-00591-f003:**
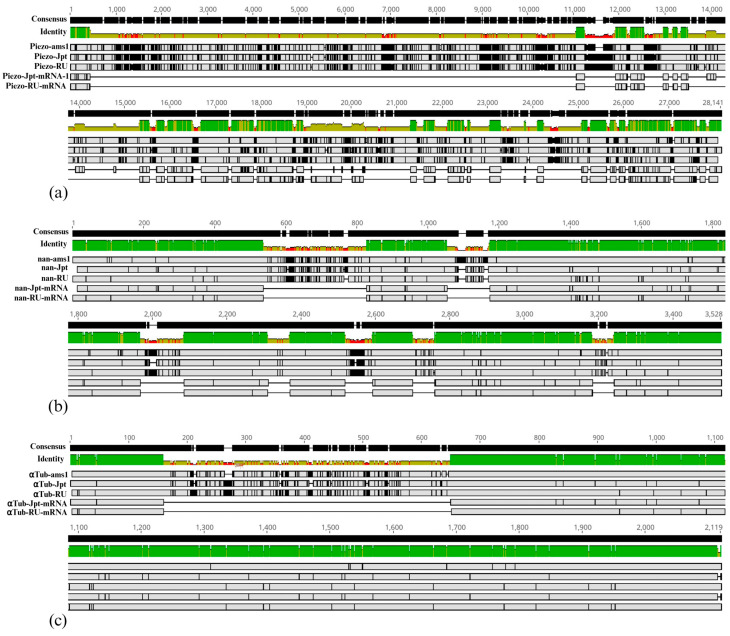
Graphical representation of multiple alignments performed by using gene and transcript nucleotide sequences corresponding to ICDPP-ams-1 (ams1), CBGP_Dsuzu_IsoJpt1.0 (Jpt), and Dsuz_RU_1.0 (RU) populations. From top to bottom, each multiple alignment provides an overview of the consensus sequence (selected coordinate values descriptive for the multiple alignment are represented above), identity fraction, with green coloring being indicated the regions containing identical nucleotides in all five sequences, the gene sequence from ams1, Jpt and RU, respectively, and the reference transcript sequences from Jpt and, lastly, RU. Regardless of the designated gene, *Piezo* (**a**), *nan* (**b**), or *αTub* (**c**), the continuous horizontal lines represented within the transcript sequences uncover the regions corresponding to the specific introns of the respective gene. The vertical black lines (representative for isolated single nucleotide variants) or rectangles (representative for successive single nucleotide variants or broader indels) from the inside of genes and transcripts visuals point to conflictual sites or subregions containing a difference in at least one sequence as compared with the other ones. Created in BioRender.com (accessed on 16 April 2025).

**Figure 4 insects-16-00591-f004:**
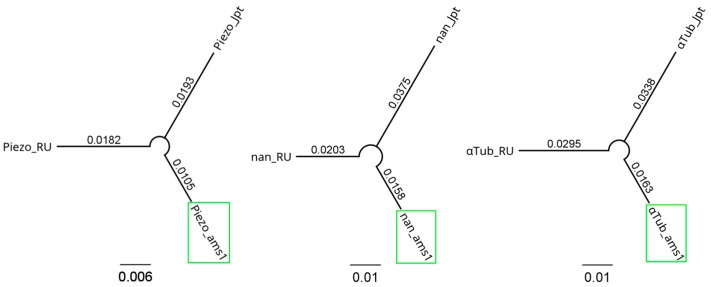
The phylogenetic trees showing the evolutionary history of *Piezo*, *nan*, and *αTub* gene structure from ICDPP-ams-1 (“gene name”_ams1), Dsuz_RU_1.0 (“gene name”_RU), and CBGP_Dsuzu_IsoJpt1.0 (“gene name”_Jpt) populations of *D. suzukii*. Regardless of the gene in question, when comparing the Romanian population with the other two, the minimum evolutionary distances are calculated between the ams1 and RU populations. The variant of each gene pertaining to ICDPP-ams-1 is highlighted with a green rectangle. Created in BioRender.com (accessed on 16 April 2025).

**Figure 5 insects-16-00591-f005:**
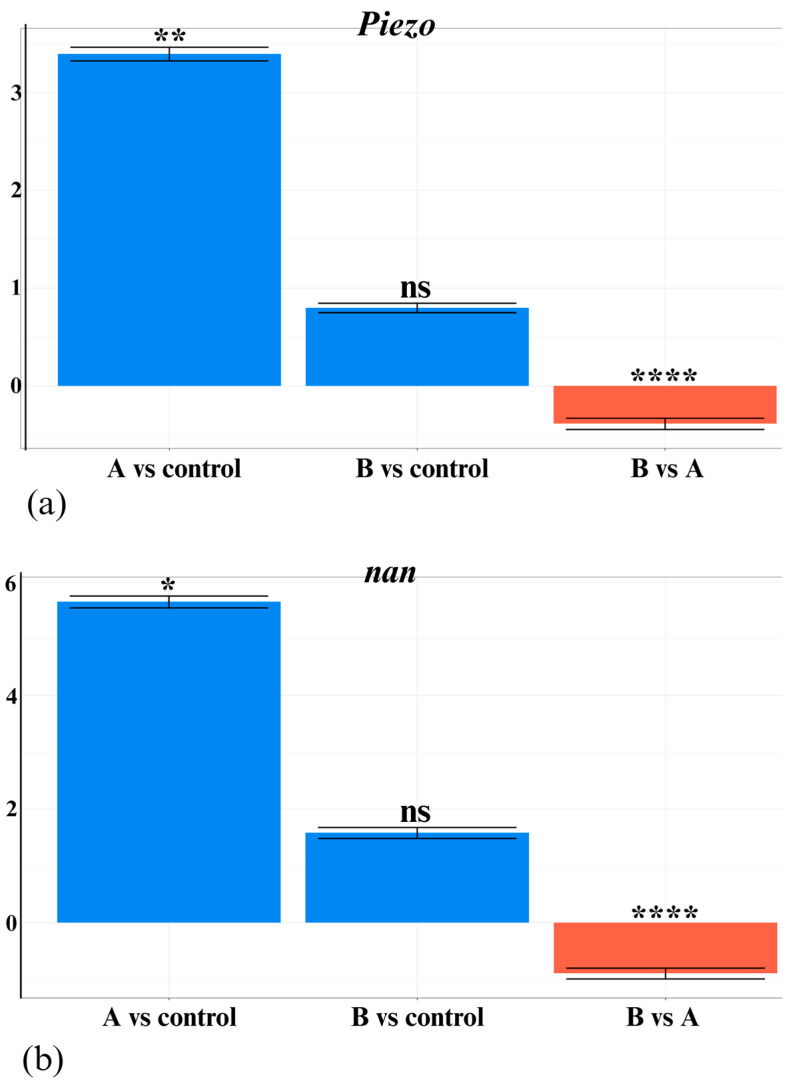
Graphical representations of log_2_FC values calculated for qRT-PCR cumulative data and statistical summary obtained by implementing the Mann–Whitney U test. Individual graphs for both *Piezo* (**a**) and *nan* (**b**) genes were generated in qDATA. Asterisks indicate statistically significant differences: *p* value < 0.05 (*), *p* value < 0.01 (**), and *p* value < 0.0001 (****). Differences labeled ns are not statistically significant. Blue bars are representative of gene overexpression, and red bars for gene downregulation. Created in BioRender.com (accessed on 16 April 2025).

**Table 1 insects-16-00591-t001:** The assessment of conflictual coordinates evidenced within the exons and introns of *Piezo*, *nan*, and *αTub* genes. Both actual counting and corresponding percentages are presented. For each gene, we calculated the intron fraction and MAT parameter. The results of Fischer’s exact test are statistically significant. In the case of *Piezo*, we presented the results corresponding either to the complete gene (all) or to a shorter variant, lacking the first 5′UTR intron (partial).

Gene	Marked Conflictual Single Coordinates			Introns Fraction	Fisher’s Exact Test	MAT
	Total (%)	Exons (%)	Introns (%)			
*Piezo* (all)	1891 (100)	195 (10.31)	1696 (89.69)	69.14%	*p* < 0.0001	1.297
*Piezo* (partial)	1230 (100)	195 (15.85)	1035 (84.15)	50.25%	*p* < 0.0001	1.675
*nan*	307 (100)	103 (33.55)	204 (66.45)	21.74%	*p* < 0.0001	3.057
*αTub*	190 (100)	58 (30.53)	132 (69.47)	22.03%	*p* < 0.0001	3.153

**Table 2 insects-16-00591-t002:** Summary of the log_2_FC values and *p* values provided by statistical analysis in the form of the Mann–Whitney U test. The FC values were calculated based on Livak’s method, while the statistical tests made use of 2^−ΔCt^ values. The calculations were performed on Ct data collected from either the first (A1 and B1) or the second (A2 and B2) experiment, respectively, as well as on their cumulative Ct data (A and B). The statistics involving the comparisons of B vs. A groups are indicated only for cumulative data.

Gene	*Piezo*			
**Experimental Group**	**A1**	**A2**	**B1**	**B2**
**Log_2_FC**	+1.54	+1.242	−1.613	+1.040
***p* value (vs. control)**	0.011	7.809 × 10^−6^	3.484 × 10^−8^	7.899 × 10^−3^
	**A**		**B**	
**Log_2_FC**	+3.339		+0.798	
***p* value (vs. control)**	1.783 × 10^−3^		0.784	
***p* value (B vs. A)**	9.106 × 10^−5^			
**Gene**	** *nan* **			
**Experimental group**	**A1**	**A2**	**B1**	**B2**
**Log_2_FC**	+2.077	+3.512	−1.974	+0.473
***p* value (vs. control)**	0.271	2.566 × 10^−6^	3.233 × 10^−8^	0.324
	**A**		**B**	
**Log_2_FC**	+5.65		+1.582	
***p* value (vs. control)**	0.012		0.168	
***p* value (B vs. A)**	2.666 × 10^−7^			

## Data Availability

Genome assembly data from NCBI: ICDPP-ams-1 (GenBank: GCA_040114545.1), Dsuz_RU_1.0 (GenBank: GCA_037355615.1), and CBGP_Dsuzu_IsoJpt1.0 (GenBank: GCA_043229965.1).

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
