# Peer review of "Short-Term Evolutionary Features and Circadian Clock-Modulated Gene Expression Analysis of Piezo, nanchung, and αTubulin at 67C in a Romanian Population of Drosophila suzukii"

_insects, 2025, doi:10.3390/insects16060591_

Round 1
Reviewer 1 Report
Comments and Suggestions for Authors
The authors want to compare evolutionary changes of DNA sequence for piezo, nanchung and a-tublin between Japan, USA and Rumanian in D Suzuki. Furthermore, they showed a different Piezo and Nanchung gene expression to alpha-tublin in Dorosophila Suzuki after 16L; 8D condition for 5 days or 3 days. They want to insist photo-periodical change of these two genes expression might be related to rapidly evolving of D Suzuki to new environment to USA and Romania from Japan.
I think the idea that TRP ion channel(nan) and mechanical vibration sensor(piezo) might be involved in evolution is very interesting. But you should revise more precisely to understand for general readers mentioned below.
- MTA; As MTA is low in piezo to nan suggest the mutation in coding of Piezo is high. What do you explain this difference of piezo to nan?
2) You should discuss different roles of piezo and nan to environmental photoperiod change .
3) You showed different expression of two genes in female fly.
It is better to show the data of male Drosophila S and discuss more deeply.
4)You should explain clearly why you checked piezo and nan expression. Did you screen other genes expression. For example, lark or other clock genes?
5)In Fig4, you should explain more clearly what you are going to suggest this phylogenic trees.
Reviewer 2 Report
Comments and Suggestions for Authors
The authors compare three isolates of Drosophila suzukii and examine the DNA sequences of three genes: Piezo, nan, and alpha Tubulin at 67 °C. They show that short-term evolutionary accumulated single-nucleotide and indel variants are overrepresented in the introns and that the housekeeping gene alpha Tubulin had the least changes. They then examined gene expression of Piezo and nan after a series of photoperiodicity challenges. Although there is some variation in the results, there is an overall trend in which Piezo and nan expression goes up with flies are shifted to long days (16 h light: 8 h dark). This trend is not observed if they are shifted to long days for 3 days and then shifted back to 12 hours light and 12 hours dark.
- It wasn’t completely clear to me why Piezo and nan were picked for this study relative to all other possible candidate genes. Additional explanation in the introduction would be useful.
- The photoperiodicity challenge experiments are interesting. Adding one more condition in which the flies are shifted to short days (8 hours light:16 hours dark) would help define the mechanism i.e., are Piezo and nan expression increasing simply from the stress of the unusual day length or is it light mediated.
- Figure 1 was very helpful in understanding the experimental design.
- Line 501—surprised should be surmised.
Reviewer 3 Report
Comments and Suggestions for Authors
In this study, Musca et al. investigated the genomic sequence of piezo and nun across species, as well as the gene expression level of them under photoperiod changes. Overall, the experiments and findings were clearly presented, and there are interesting findings. However, certain improvements can be made to make it more solid and be of greater interest to the audience.
1.My major concern is that how do the two approaches - Genomic sequence phylogenetic analysis and gene expression analysis - align with each other to conclude “short-term evolution”. There are several reasons for it:
- The phylogenetic analysis did show difference between the 3 populations, and as the authors stated, more differences in intron than in exon, and heavier intron fraction in Tub than in Piezo and nun. However, if these are compared with other Drosophila species, it would better show a “short-term” evolution, as it will be known whether this is different from longer term evolution. Also, even though “short-term evolution” is both in the title and the keyword, the introduction and discussion about the specificity of it is not enough.
- How is the gene-expression level under acute photoperiod changes related to evolution? It is not even photoperiod changes across generations; it looks more like an acute and plastic change. If the authors speculate that the gene expression level changes eventually will lead to trait change, please describe that and provide references.
- For qpcr, please use at least one more housekeeping gene for normalization, and compare the gene expression level of Tub to it.
- Why did the authors focus on photoperiod changes in Romania population? Is the photoperiod or light condition very different in Romania compared to Japan and USA? Please discuss the rationale.
2.In methods part, please list the qpcr primer sequence for Tub, nan, and Piezo
3.To make the paper easier to read, in the result sections, line 317 to 322, please translate the labels of “ICDPP-ams-1” “Dsuz_RU_1.0””CBGP_Dsuzu_IsoJpt1.0” to more human language such as “Romania local population””USA population” and “Japan population”, and discuss how to interprate these results in the context of geographical differences and evolutionary time of these three populations.
4.Similarly, in section 3.2, Table 2, and Fig. 5, please replace “A” “B” with some words that makes more sense. It is really hard for the readers to go back and forth to check what is A and B.
Round 2
Reviewer 1 Report
Comments and Suggestions for Authors I am agree to accept the revised MS.